# Learning to Find Proofs and Theorems by Learning to Refine Search Strategies

## The Case of Loop Invariant Synthesis

**Jonathan Laurent**
Carnegie Mellon University
Karlsruhe Institute of Technology

**André Platzer**
Carnegie Mellon University
Karlsruhe Institute of Technology

## Abstract

We propose a new approach to automated theorem proving where an AlphaZero-style agent is self-training to refine a generic high-level expert strategy expressed as a nondeterministic program. An analogous teacher agent is self-training to generate tasks of suitable relevance and difficulty for the learner. This allows leveraging minimal amounts of domain knowledge to tackle problems for which training data is unavailable or hard to synthesize. As a specific illustration, we consider loop invariant synthesis for imperative programs and use neural networks to refine both the teacher and solver strategies.

## 1   Introduction

Augmenting tactic-based interactive theorem provers with neural guidance has been the focus of increased attention in recent years [1, 2, 3, 4, 5]. The dominant approach uses imitation learning on corpora of formalized mathematics. However, despite recent efforts involving self-supervised pre-training [5] or data-augmentation [6], this approach is limited by the conspicuous scarcity of human-produced training data. An alternative approach inspired by the success of AlphaZero [7] is to use reinforcement learning and let an agent self-train to interact with a theorem prover via trial and error [8, 9]. However, previous attempts of doing so have been hampered by two fundamental issues:

- First, existing tactic-based theorem provers offer infinite action spaces not amenable to random exploration. Also, they are optimized for formalizing the outcome of human insights concisely but often fail to define a tractable search space for deriving those insights in the first place [10]. For example, using tactics effectively often requires providing insights in advance that are more easily found later in the search process (e.g. constructing a term to instantiate an existential quantifier [11]). More generally, most of the human process of discovering proofs, which involves trial and error, sketching, and abductive reasoning is not captured by standard prover tactics and so we can hardly expect a reinforcement learning agent to learn this process through sheer interaction with a tactic-based prover.

- Second, although reinforcement learning alleviates the need for human proofs during training, the agent must still be provided learning tasks of suitable relevance, diversity, difficulty, and generalizability in the form of theorem statements to be proved. This is in contrast with applications of AlphaZero in board games, where the symmetric nature of games such as Chess or Go enables leveraging symmetric self-play as an infinite source of training tasks.

In this paper, we suggest a novel approach to learning automated theorem proving that does not rely on human-provided proofs and theorems. Instead, a teacher agent is trained to generate interesting and relevant tasks while a solver agent is co-trained to solve them. Our core insight is that *both* agents can be implemented by using reinforcement learning to refine high-level search *strategies* [10] written by

36th Conference on Neural Information Processing Systems (NeurIPS 2022).

```
assume x >= 1;
y = 0;
while (y < 1000) {
  x = x + y;
  y = y + 1;
}
assert x >= y;
```

A desirable *invariant* is a formula $I[x, y]$ such that:

$$\forall x, y \begin{cases} x \geq 1 \wedge y = 0 \; \rightarrow \; I[x, y] & \text{(holds initially)} \\ y < 1000 \wedge I[x, y] \; \rightarrow \; I[x + y, y + 1] & \text{(preserved)} \\ y \geq 1000 \wedge I[x, y] \; \rightarrow \; x \geq y & \text{(implies post)} \end{cases}$$

Such an invariant is $I[x, y] := (x \geq y \wedge x \geq 1 \wedge y \geq 0)$

Figure 1: An example program and an associated loop invariant

experts in the form of nondeterministic programs. Our paper also defines novel design principles and language constructs for expressing such strategies. These include *conditional generative strategies* as a template for implementing teachers, *abductive reasoning* as a design principle that is made scalable by self-learned guidance, and *strategy events* as an abstraction that enables easier reward engineering and better sample-efficiency. This paper introduces our new framework and also evaluates it in a well-contained yet challenging setting, namely the automatic verification of imperative programs with loops. We frame our contributions in a way to emphasize general applicability. An empirical evaluation in other application domains is left to future work.

## 2 Approach

### 2.1 Background: Verifying Imperative Programs with Loops

Suppose we are given a program such as the one in Figure 1 and we want to prove that the final assertion always holds. The way to proceed is to find a predicate called a *loop invariant* with the following properties: *i)* it is true before the loop is entered, *ii)* it is preserved by the loop body when the loop guard holds and *iii)* it implies the final assertion when the loop guard does not hold. By "preserved", we mean that if the invariant is true before executing the loop body then it is also true afterwards. Finding loop invariants is the most crucial aspect of program verification [12, 13] and still resists automation [14].

Despite the difficulty of automatically synthesizing loop invariants, humans do so routinely using well-understood search strategies. For example, in the case of the example in Figure 1, one would first try to prove the postcondition $x \geq y$ itself as an invariant and then note that it is not preserved by the loop body as the following implication does not hold: $y < 1000 \wedge x \geq y \; \nrightarrow \; x + y \geq y + 1$. However, simplifying the right-hand side, we see that the proof works if we can show $x \geq 1$ to be an invariant itself. Using a similar form of abductive reasoning, we find that this in turn requires $y \geq 0$ to be an invariant and we end up proposing $x \geq y \wedge x \geq 1 \wedge y \geq 0$ as a loop invariant.

### 2.2 Expressing Search Strategies as Nondeterministic Programs

The search strategy we followed for the example above can be formalized as a nondeterministic program, which is shown in Figure 2. As a nondeterministic program, it features a `choose` operator that takes a list of objects as an input and selects one of them nondeterministically. A nondeterministic program can be refined into a deterministic one by providing an external oracle to implement the `choose` operator. It also defines a search tree that can be explored with or without a guiding heuristic. In this work, we use neural networks to implement such oracles and guiding heuristics.

In addition, a key aspect of this strategy is to infer missing assumptions under which a currently failing proof obligation would hold. This form of reasoning is called *abductive reasoning* and it is fundamental in the way humans search for proofs [16, 17]. It is implemented in a separate `abduct` procedure that takes a formula as an input and returns either `Valid` or a list of abduction candidates. For example, the `abduct` procedure fails to prove the implication $x \geq 0 \rightarrow x + y \geq 1$ but may suggest $x + y \geq 1$, $x < 0$ and $y \geq 1$ as possible missing assumptions. The use of abductive reasoning for theorem proving and loop invariant synthesis specifically has been proposed in the past [18, 19]. However, abduction procedures are hard to implement [20] and typically only available for specific decidable theories. Also, using them in proof search tends to scale poorly in the absence of

```
1   def solver(
2       init: Formula, guard: Formula,
3       body: Program, post: Formula) -> Formula:
4
5     def prove_inv(inv: Formula) -> List[Formula]:
6       assert valid(Implies(init, inv))
7       inductive = Implies(And(guard, inv), wlp(body, inv))
8       match abduct(inductive):
9         case Valid:
10          return [inv]
11        case [*suggestions]:
12          aux = choose(suggestions)
13          return [inv] + prove_inv(aux)
14
15    inv_cand = choose(abduct(Implies(Not(guard), post)))
16    inv_conjuncts = prove_inv(inv_cand)
17    reward(max(-1, -0.2 * len(inv_conjuncts)))
18    return And(*inv_conjuncts)
```

Figure 2: A simple strategy for finding loop invariants, described in Python-like pseudocode syntax. In this strategy, an initial invariant candidate is selected nondeterministically that implies the postcondition (line 15). One ensures that this candidate holds initially (line 6), without which the strategy fails immediately. Then, one attempts to prove that it is preserved by the loop body (lines 7 and 8, where wlp denotes Hoare's weakest liberal precondition operator [15]). If the candidate is not preserved, the abduct function suggests a list of assumptions that make it so. One then uses the choose operator to nondeterministically select one of them, which we try and prove invariant recursively (line 13). Because the number of abduction candidates to choose from can be large, successfully using such a strategy depends on having an effective oracle to guide search. To provide sufficient context to such an oracle, calls to choose must provide some extra information that is omitted here for brevity. In this case, we would pass a special token to indicate the call site, the program being analyzed and the value of inv in the case of line 12. See Appendix A for more details and for a full listing of the invariant synthesis strategy that we use in our experiments.

good heuristics to rank and filter abduction candidates. Our proposed framework makes abductive reasoning practical by addressing both issues: it allows leveraging self-learned guidance to select abduction candidates and it allows implementing abduction procedures as nondeterministic programs that are amenable to learning themselves.

Another notable feature of the strategy in Figure 2 is the use of the reward operator on line 17 to incentivize finding short invariants (short proofs are desirable in general as they tend to be easier to check, interpret and generalize). The reward operator can be used multiple times and at any point of the strategy execution. An implicit reward of 1 or -1 is emitted when the strategy successfully returns or fails respectively. An optimal execution of a nondeterministic strategy is one that maximizes the cumulative amount of collected rewards. In this example, we bound the maximal invariant size penalty in such a way to ensure that finding a proof is always rewarded more than failing at doing so.

As argued by Selsam [10], defining search strategies as nondeterministic programs provides a natural and flexible way for experts to leverage domain-specific knowledge while outsourcing difficult proof decisions to search algorithms and learned heuristics. This paradigm differs from the standard paradigm of tactic-based theorem provers in which an external entity must orchestrate stateless tactics that do not call for any form of interactive guidance internally. In contrast, the nondeterministic programming approach *inverts* control and has strategies call external oracles rather than the other way around, which allows for a much tighter control of the resulting search space.

## 2.3   Refining Nondeterministic Strategies with Reinforcement Learning

A nondeterministic program induces a (deterministic) Markov Decision Process (MDP) where intermediate states are choice points and final states are either success states (the program returns successfully) or failure states (an assertion is violated or choose is called on an empty list of choices).

Standard search algorithms can be used to navigate this MDP but doing so in an efficient and scalable way requires strong heuristics for guiding search.

In this work, we propose to learn such heuristics in a self-supervised fashion using the AlphaZero algorithm [7, 21]. In AlphaZero, a neural network is trained to map any state to a probability distribution over available actions along with a value estimate (i.e. an estimate of the expected sum of future rewards). The neural network alone can be used as a policy for navigating the MDP. However, a stronger policy results from combining the network with a tree search algorithm such as Monte-Carlo Tree Search (MCTS) [22]. The key insight underlying AlphaZero is that the network can be trained via an iterative improvement process where it is successively *i)* used as an MCTS heuristic to try and solve problems and then *ii)* updated using gradient descent so as to better predict the outcome of each attempt along with the action selected by MCTS on all states encountered along the way. More details on AlphaZero can be found in the literature [7].

Using AlphaZero, we can refine nondeterministic proof search strategies *without* external proof examples to learn from. However, AlphaZero must still be provided with a set of training problems to be solved. Training problems should be diverse, relevant and numerous enough to allow proper generalization. They should also be of varied difficulty to allow for learning to bootstrap.

## 2.4 Generating Training Problems with Conditional Generative Strategies

In theorem proving, high-quality training problems produced by humans are often not available in quantities even remotely matching the needs of reinforcement learning to properly generalize across problem instances. A possible solution is to use procedural generation techniques to assemble large datasets of synthetic problems. This works well in some specific areas [23] and particularly for problems that can be framed as inverse problems such as symbolic integration [24]. In other areas, generating interesting problems is as hard as solving existing ones, possibly harder.

This is the case in particular with the problem of loop invariant synthesis. Here, a natural idea for generating problem instances would be to use a probabilistic grammar for repeatedly sampling triples consisting of a program, a loop invariant and an assertion and then reject all triples in which the properties defining valid invariants do not hold. Unfortunately, not only would doing so naively lead to a very high rejection rate, but the resulting dataset would be heavily biased towards trivial samples that are hardest to reject (e.g. programs where the final assertion is the negation of the loop guard and where the invariant does not even matter). In contrast, many classes of interesting problems from standard human-written benchmarks would only be sampled with an infinitesimal probability.

In this work, we propose to generate problems using the same methodology we use to solve them: by leveraging reinforcement learning to refine nondeterministic search strategies.

Specifically, experts can define *teacher strategies* in the form of stochastic and nondeterministic programs, each run of which either fails or successfully returns with a problem instance. Reinforcement learning can then be used to refine such programs with the two objectives of maximizing the diversity and interestingness of generated problems while avoiding rejection. However, these objectives are naturally in tension since an agent may be locally incentivized to avoid generating particular types of problems that are easier to reject. We introduce a design template for a class of strategies that avoid this obstacle. We call such strategies *conditional generative strategies*.

A conditional generative strategy generates a problem in three steps: *i)* it samples a set of constraints that define desired features for the generated problem, *ii)* it nondeterministically generates a problem that respects as many constraints as possible and gets rewarded for doing so and *iii)* it applies a sequence of random and validity-preserving problem transformations as a way to further increase diversity. We provide a high-level description of a such a teacher strategy for invariant synthesis in Figure 3. The interest of such an architecture is that it enables decoupling the two conflicting objectives of generating valid and diverse problems: diversity is guaranteed by the first and third steps while learning can focus on the second step. This also allows using the exact same learning algorithm for training both teacher and solver agents (AlphaZero in this work).

## 2.5 Predicting Events Rather Than Rewards

In the standard AlphaZero setting, the network is trained to predict the value of encountered states, that is the expected sum of future rewards. However, such a number conflates a lot of valuable

```
1   def teacher(rng: RandGen) -> Prog:        19      p = refine_inv(p, cs)
2       cs = sample_constrs(rng)              20      p = refine_body(p, cs)
3       p = generate_prog(cs)                 21      assert valid(inv_preserved(p))
4       p = transform(p, rng)                 22      p = refine_post(p, cs)
5       p = hide_invariants(p)                23      assert valid(inv_post(p))
6       return p                              24      p = refine_init(p, cs)
7                                             25      assert valid(inv_init(p))
8   def generate_prog(cs: Constrs):           26      nv = num_violations(p, cs)
9       p = Prog("                            27      reward(max(-1.5, -0.5 * nv))
10          assume init;                      28      return p
11          while (guard) {                   29
12            invariant inv_lin;              30  def transform(p: Prog, rng: RandGen):
13            invariant inv_aux;              31      p = shuffle_formulas(p, rng)
14            invariant inv_main;             32      p = shuffle_instrs(p, rng)
15            body;                           33      p = add_useless_init(p, rng)
16          }                                 34      p = add_useless_post(p, rng)
17          assert post;")                    35      ...
18       p = refine_guard(p, cs)              36      return p
```

Figure 3: Simplified teacher strategy. Problems are generated by nondeterministically refining the template in lines 10-17 in a way to optimize random constraints (see examples in Table 1). In this template, the invariant is expressed as a conjunction of at most three sub-invariants: `inv_lin` is a linear equality or inequality, `inv_main` is a disjunction of atomic formulas (i.e. comparisons) and `inv_aux` is a conjunction of atomic formulas that can be used in proving `inv_main` but not `post`. These three formulas are refined nondeterministically by the call to `refine_inv` on line 19. After an invariant is chosen, a loop body is also generated nondeterministically (line 20) that preserves the invariant (line 21). Finally, the `num_violations` function penalizes the presence of useless or redundant problem components. For example, a penalty is applied if `inv_main` features a disjunct that can be removed without invalidating the problem. Appendix B provides more details on how the `refine_*` functions can be implemented. The simplest way to do so is to have a grammar and use the `choose` operator to select rules recursively. However, fancier strategies can easily be implemented in the nondeterministic programming framework. In particular, our implementation uses the `abduct` procedure introduced in Figure 2 to suggest values for constants and subformulas.

| Name | Type | Description |
|---|---|---|
| `num-inv-main-disjuncts` | none\|1\|2 | If an integer $n$, then `inv_main` is refined with a disjunction of $n$ atomic formulas. |
| `has-conditional` | bool | Whether body must include a conditional statement. |
| `loop-guard-useful-for-post` | bool | Whether assuming the negation of the loop guard is useful in proving the postcondition `post`. |
| `body-implies-main-inv` | bool | If `true`, then `inv_main` always holds after executing body regardless of whether or not it holds before. |
| `eq-only-for-init` | bool | Whether or not to use equalities only in `init`. |

Table 1: Examples of teacher constraints. Some descriptions refer to names introduced in Figure 3. The full list of constraints we used in our experiments is available in Appendix B.

information. For example, suppose a teacher state is assigned a value of 0. Does 0 mean that the teacher is predicted to either fail or succeed with equal probability or that it will certainly succeed in generating a problem that violates two constraints? And if so, which two?

Although this information is not relevant to MCTS, there are several reasons for having the network predict it anyway. First, doing so can result in an increased sample efficiency by providing the network with more detailed feedback on its mistakes. Such an effect has been previously demonstrated in the KataGo project [25]. Second, it is interesting from an interpretability point of view and makes it easier to diagnose problems and weaknesses in a given network. We found a third reason that is more subtle, which has to do with getting the combined benefits of issuing intermediate and final rewards.

Indeed, in the teacher strategy showed in Figure 3, we only penalize the agent for violating a constraint *at the very end*. Intuitively, there is a lost opportunity here since many violations could be detected much earlier while the problem is still being refined. Emitting an intermediate reward at this point would certainly improve learning efficiency. However, the same violation can be potentially detected at several different points in time and we must ensure that a reward is only issued the first time. This in turns makes the job of the value prediction network harder since it needs to keep track of whether or not each violation has been penalized already. Encoding this information alone can be costly, and especially so for attention-based network architectures such as transformers with a quadratic inference cost. Having the network predict constraint violations separately offers an elegant way out: all rewards are issued at the end but from the moment a constraint violation is detected, the network's predicted probability for this violation is overridden to 1 whenever a value estimate is computed.

More formally, we propose to introduce the concept of an *event*. Strategies can declare an arbitrary number of events $e$ and each one is associated with a reward $r_e$ along with a maximal number of occurrences $m_e$ ($m_e = 1$ for constraint violation events) that can be counted towards the final reward. The `reward` operator introduced in Figure 2 is replaced by an `event` operator. When a strategy terminates, a reward is issued implicitly with a value of $-1$ in case of a failure and

$$\max\left\{1 + \sum_e r_e \min(n_e, m_e),\ r_{\min}\right\}$$

in case of a success. In this expression, $n_e$ denotes the number of calls made to `event`$(e)$ during the whole episode and setting $r_{\min} > -1$ guarantees that successes are always rewarded more than failures. Note that it is important not to issue event penalties after a failure. Otherwise, faced with a likely failure, an agent may be incentivized to simply give up and fail immediately to avoid further penalties in search of success. Also, there are two reasons for bounding the maximal number of occurrences of event $e$ that are counted towards the final reward by a constant quantity $m_e$. First, this allows having a network with a finite softmax head predict the number of occurrences of $e$ (as we see below). Second, the $m_e = 1$ case is particularly useful to model events such as constraint violations that can be detected multiple times but should only be penalized once.

In this framework, the value head of the network is tasked with predicting the following probabilities: *i)* the probability $p_0$ of a failure, *ii)* the probability $p_1$ of succeeding with minimal reward $r_{\min}$, *iii)* the probability $p_2 = 1 - p_0 - p_1$ of succeeding with a greater reward and *iv)* the probabilities $p_e^i$ that $\min(n_e, m_e) = i$ for all events $e$ and $0 \leq i \leq m_e$. A value estimate derives from these probabilities:

$$-p_0 + p_1 \cdot r_{\min} + p_2 \cdot \sum_e \sum_i \hat{p}_e^i \cdot i \cdot r_e$$

where $\hat{p}_e^i \propto \mathbf{1}\{n_e \leq i\} \cdot p_e^i$ is a corrected probability estimate accounting for the number of times $n_e$ that `event`$(e)$ was raised already. In our invariant synthesis experiments, we use events in both the teacher and solver strategies to represent constraint violations ($m_e = 1$) and encourage short proofs by penalizing individual proof steps respectively ($m_e > 1$).

## 3  Implementation and Engineering Contributions

This paper advocates for a hybrid approach to automated theorem proving where human experts from a variety of areas can formalize their knowledge succinctly in the form of nondeterministic proving and teaching strategies. Reinforcement learning is leveraged to fill in the blanks in all inevitable cases where the human intuition escapes formalization.

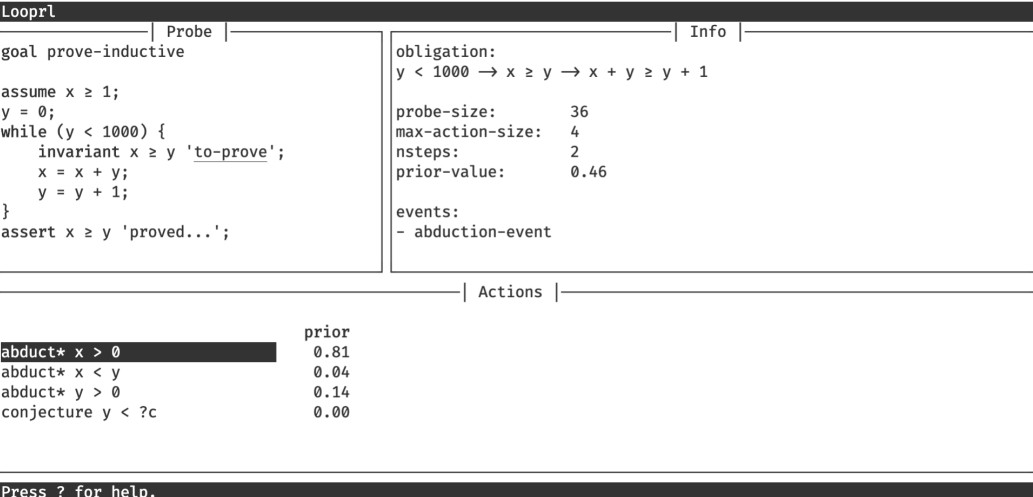

```
┌ Looprl ────────────────────────────────────────────────────────────────────┐
│ ┌─────────────────┤ Probe ├──────┐   ┌──────────────────┤ Info ├──────────┐ │
│ │ goal prove-inductive           │   │ obligation:                       │ │
│ │                                │   │ y < 1000 → x ≥ y → x + y ≥ y + 1   │ │
│ │ assume x ≥ 1;                  │   │                                   │ │
│ │ y = 0;                         │   │ probe-size:        36             │ │
│ │ while (y < 1000) {             │   │ max-action-size:   4              │ │
│ │     invariant x ≥ y 'to-prove';│   │ nsteps:            2              │ │
│ │     x = x + y;                 │   │ prior-value:       0.46           │ │
│ │     y = y + 1;                 │   │                                   │ │
│ │ }                              │   │ events:                           │ │
│ │ assert x ≥ y 'proved...';      │   │ - abduction-event                 │ │
│ └────────────────────────────────┘   └───────────────────────────────────┘ │
│                          ┌──────────┤ Actions ├──────────────────────────┐  │
│                          │                prior                          │  │
│                          │ abduct* x > 0    0.81                         │  │
│                          │ abduct* x < y    0.04                         │  │
│                          │ abduct* y > 0    0.14                         │  │
│                          │ conjecture y < ?c 0.00                        │  │
│                          └───────────────────────────────────────────────┘  │
│ Press ? for help.                                                           │
└─────────────────────────────────────────────────────────────────────────────┘
```

Figure 4: Using the Looprl UI to inspect the solver agent on the problem from Figure 1. Additional commented screenshots are available in Appendix E.

However, for this vision to be practical, writing strategies and iterating on them must be as frictionless as possible. New tools and abstractions are needed to automate the process of converting nondeterministic programs into reinforcement learning environments, ease debugging and foster code reuse. In this work, we accomplish a crucial step in this direction by introducing the Looprl theorem prover. Looprl consists of the following collection of features:

- A domain specific language for writing strategies embedded in OCaml, with first-class support for neural-guided nondeterministic programming. Strategies are automatically compiled into reinforcement learning environments that can be explored in Python.

- A graphical interface that can be used to interact with strategies manually while inspecting the neural network's predictions and the behavior of MCTS (see Figure 4).

- A library of utility functions for writing program verification strategies. This library includes an implementation of the `abduct` function introduced in Figure 2 for integer linear arithmetic (see details in Appendix F).

- A parallel, high-performance implementation of the (Gumbel) AlphaZero algorithm [7, 21] for refining search strategies. Our implementation uses Pytorch [26] and Ray [27]. It provides support for events (see Section 2.5) and unbounded, variable-size action spaces.

The first point on developing a domain-specific language for writing strategies is particularly important and deserves more discussion. In our experience, having such a language is not just a mere matter of convenience but one of feasibility. In fact, we did try to implement an initial version of a teacher strategy for loop invariant synthesis by writing pseudocode on paper and manually compiling it into an MDP implemented in Python. However, doing so resulted in a complexity explosion where any single-line change to the pseudocode could take days of work to implement.

The reason compiling a nondeterministic program into an MDP manually is so tedious is that the whole program state must be made explicit, which includes the program stack. A tempting alternative would be to write nondeterministic code as normal Python code parametrized by an arbitrary `choose` function. However, this does not allow using tree search on the resulting program since doing so would require cloning the whole execution state of a Python program. Fortunately, there exists a solution to this problem in the field of programming languages using *search monads* [28, 10] and which essentially enables writing arbitrary nondeterministic code and then reifying it into a search tree that can be manipulated explicitly.

In Looprl, we implement a domain specific language embedded in OCaml for expressing nondeterministic strategies. OCaml is a natural choice because it is fast (about two orders of magnitude faster than Python for symbolic code such as proof inferences) and it has good support for monads and Python interoperability. Our language provides built-in support for the `choose` and `event` operators.

Also, it defines an intermediate graph-based format for representing data to be sent to neural networks in an architecture-agnostic way. Any piece of information that is passed to the `choose` operator must be convertible to this format and feature suitable metadata ensuring a seamless integration with the Looprl user interface and debugging tools.

## 4   Experiments

We tested Looprl on the problem of synthesizing loop invariants for single-loop imperative programs and evaluated it on the standard Code2Inv [29] benchmark. This benchmark contains a set of 132 programs written in C. All programs feature a single loop along with a final assertion to be proven. They feature linear integer arithmetic and sometimes involve conditionals, assumptions and nondeterministic tests (some Code2Inv problems are in Appendix C). Most existing invariant generation tools can only solve a subset of these problems [30]. To the best of our knowledge, only one existing tool can solve them all [31]. However, it does not use learning and relies on specific optimization techniques that are not generalizable beyond the setting of purely numerical programs.

We used Looprl to implement an invariant generation strategy, which we describe in Appendix A. In a nutshell, it generalizes the simple strategy described in Figure 2 by adding features such as the ability to abduct disjunctive invariants or to conjecture invariant templates with placeholder constants to be refined later using abduction. To our surprise and although this generic strategy can be described in only one page of pseudocode, it is sufficient to simply solve *every* Code2Inv benchmark problem in a few seconds when combined with the vanilla MCTS algorithm [22] (with no learned heuristic).

We therefore evaluate a self-trained solver agent on its ability to solve as many Code2Inv problems as possible *without* resorting to any search. At every decision point, it greedily takes the highest ranked action according to the network's policy head and no backtracking is allowed. We argue that such a metric is of particular interest and significance. Indeed, finding an invariant for a single loop is of limited interest of its own. Rather, doing so is typically useful as a subtask in a hierarchy of increasingly complex and relevant problems. The next level of this hierarchy may be to prove a piece of code with nested loops, and then to synthesize a function respecting a specification, which is still several levels away from writing full-fledged software systems. Complex problems cannot be solved if backtracking search is needed at every level of this deep hierarchy.

### 4.1   Training Protocol

We train a teacher agent and a solver agent in sequence using the (Gumbel) AlphaZero algorithm. Both agents use a similar 1.6M parameters Dynamic Graph Transformer network [32] (see architecture details in Appendix H). The teacher agent is trained first for 20 AlphaZero iterations. During each iteration, it goes through 8000 problem generation episodes, using 64 MCTS simulations per step. Then, the network is updated using samples from the $k$ previous iterations with $k$ increasing from 1 to 6 during training. Each iteration simulates 800 additional episodes to generate *validation* samples that are not used to train the network but to control overfitting via early stopping. After the teacher agent is trained, a dataset of 50K problems is gathered for training the solver, which includes a mix of 40K problems generated during the last five training iterations and 10K new ones that are generated without Gumbel exploration noise [21]. The solver agent is also trained for 20 iterations. During each iteration, it attempts to solve 20K randomly selected problems using 32 MCTS simulations per step. In addition, 5K additional problems are solved to generate validation samples. See Appendix G for a complete list and explanation of all training hyperparameters.

A complete training run takes about 16h on a single machine with a 10-core Intel i9-10900KF processor, 64GB of RAM and a NVIDIA GeForce RTX 3080 GPU.

### 4.2   Experimental Results

We show training curves for the solver and teacher agents in Figures 5 and 6 respectively. These curves record the evolution of the average reward collected by MCTS combined with the latest network during each iteration. Error intervals are estimated based on two random seeds. Note that the maximum theoretical reward obtainable by the teacher agent is $<1$ since problem generation constraints may be sampled that are mutually incompatible, in which case the agent simply does its best to minimize the number of violated constraints. The same holds for the solver agent since all

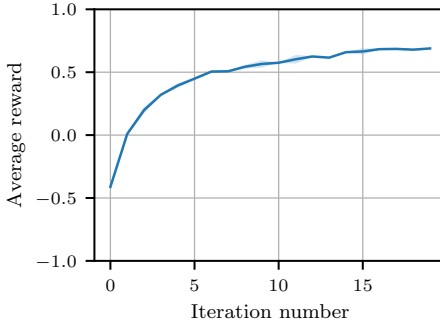

Figure 5: Teacher training curve

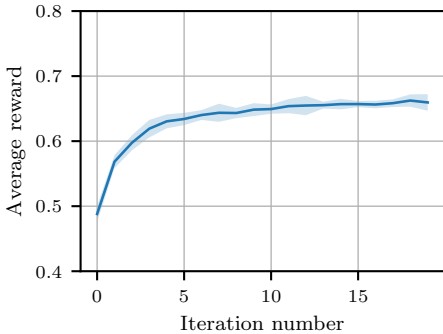

Figure 6: Solver training curve

| Policy | % Problems solved |
|---|---|
| Random | $18.4 \pm 0.0$ |
| Network (untrained teacher) | $39.7 \pm 1.6$ |
| Network (trained teacher) | $\mathbf{61.5 \pm 0.4}$ |

Table 2: Experimental results on the Code2Inv benchmark (using no backtracking search). The score for the Random heuristic is computed as an average across 10K attempts. Standard deviations are computed based on using two different random seeds for the full training experiment.

generated invariants are penalized proportionally to their size. The solver agent goes through a more modest reward gain during training (the $y$-axis is rescaled in Figure 6 to emphasize the trend). This is reflective of the fact that most problems generated by the teacher can be solved by MCTS alone and so most of the training concentrates on learning to solve about 10% of hard problems.

We show the results of our evaluation on the Code2Inv benchmark in Table 2. Three different agents are compared on the average ratio of benchmark problems for which they can successfully generate an invariant of minimal size *without* search or backtracking. The first agent is a baseline that simply selects proof actions at random. The second agent is a solver network that was trained using problems generated by an untrained teacher (i.e. using MCTS but no network heuristic). Finally, the third agent is a solver network trained using the full protocol of Section 4.1. These results demonstrate the critical impact of learning for both the teacher and solver agents. Unsurprisingly, an inferior teacher leads to an inferior solver with decreased generalization capabilities.

## 5   Related Work

**Leveraging nondeterministic programming for proof search.**   The idea of combining nondeterministic expert strategies with neural oracles was proposed by Selsam [10, 33] as a possible angle for tackling the IMO Grand Challenge [34]. However, how to train such oracles remains an open problem. One proposal [32] is to use supervised learning for training a universal oracle that is conditioned on the description of arbitrary search problems and can therefore use training data from a large number of heterogeneous sources. In contrast, we propose using reinforcement learning with the insight that nondeterministic search strategies can be used to *generate* problems in addition to solving them.

**Learning to generate synthetic theorems.**   Procedural generation techniques have been used for producing synthetic theorems [35, 36, 37, 24, 38]. These usually proceed in a backwards fashion by generating random proof trees from a given set of axioms and rules. Wang *et al.* proposed to use supervised learning to learn how to guide this process towards generating more interesting theorems that are similar to those of a reference dataset [6].

**Using reinforcement learning for loop invariant synthesis.**   Reinforcement learning has already been applied to the problem of loop invariant synthesis [29, 30], albeit in a very different way. In the aforementioned work (which introduces the Code2Inv benchmark), a *separate* reinforcement

learning agent is trained from scratch on every benchmark problem, using counterexamples from an SMT solver as a training signal. This process takes minutes to hours for each problem to be solved. In contrast, we train a *single* agent to generalize across problem instances so that new problems can be solved in a matter of milliseconds.

**Using reinforcement learning for theorem proving.**    The HOList Zero [8] and TacticZero [9] systems use reinforcement learning to learn how to interact with tactic-based theorem provers without relying on human proofs. However, they still rely on large human-produced corpora of formalized mathematical statements to be used as training tasks and are not yet competitive with approaches based on imitation learning. The use of reinforcement learning has also been explored in the context of saturation-based or tableau-based provers for first-order logic [39, 40]. However, learned heuristics in such settings must operate at a very low-level and are subject to a speed-accuracy tradeoff that is unfavorable to deep learning.

## 6    Conclusions and Future Work

This paper took a bold stride toward the challenge of self-learning automated theorem proving without relying on example theorems and proofs. It advocates a hybrid approach to theorem proving where experts are provided a flexible language to formalize their domain-specific knowledge in the form of generic nondeterministic teacher and solver strategies, leaving blanks to be filled by learning.

We demonstrated our framework by applying it to the problem of loop invariant synthesis. Our agent solves all Code2Inv challenges. More importantly, it learns to solve a majority of problems with no search at all despite never seeing these problems during training.

None of our core contributions, however, are specific to invariant synthesis. These include: *i)* our key insight that *nondeterministic programming* and *reinforcement learning* can be similarly combined to implement solvers and teachers, *ii) conditional generative strategies* as a general template to write teachers, *iii) abductive reasoning* as a design principle that is made scalable by self-learned guidance, *iv) strategy events* for easier reward engineering and better sample-efficiency, and *v)* an implementation that establishes engineering foundations for making the whole approach practical.

We see this paper as a first step toward a broader vision where interactive theorem provers allow users to write teacher and solver strategies for domains of mathematics and programming in a modular and distributed fashion. A unique neural network can then be trained as an oracle for all these strategies, allowing better generalization and sample efficiency. However, many challenges lie ahead. In particular, while there is some evidence already [10, 41, 19] that effective solver strategies can be written at scale (interactive theorem provers already allow specifying custom search tactics [42, 43, 44] and we are merely suggesting a generalization of this popular feature), scaling up teacher strategies raises more questions. For example, writing conditional generative strategies can be engineering-intensive since diversity is enforced extrinsically via manually-defined constraints. Future work will explore teachers that also integrate intrinsic reward signals, either via a curiosity mechanism or by having the solver directly reward the teacher for producing problems of suitable novelty and difficulty.

## Broader Impact

Automated theorem proving for program verification and program synthesis plays an important role for increasing the safety of critical pieces of software. More automatic theorem provers enable a wider user base to benefit from the advances to their software quality. But automating theorem provers is a highly sophisticated challenge. Learning from succinct expert advice in the form of strategies may be the sweet spot enabling more generic domain-specific automation for theorem provers.

## Acknowledgments and Disclosure of Funding

This work was supported by the Alexander von Humboldt Professorship program, the National Science Foundation under Award CNS-1739629 and by DARPA under Contract No. FA8750-18-C-0092. We thank Ruben Martins, Paul Gölz, Vincent Hellendoorn, Léonard Blier along with the anonymous reviewers at NeurIPS 2022 for their helpful comments and feedback.

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
