# OpenReview forum: "Learning to Find Proofs and Theorems by Learning to Refine Search Strategies: The Case of Loop Invariant Synthesis"
_NeurIPS.cc/2022/Conference — NeurIPS 2022 Accept_

### Official Review · Reviewer_xxib · 2022-07-11

**Rating:** 5
**Confidence:** 4
**Soundness:** 3 good
**Presentation:** 3 good
**Contribution:** 2 fair

**Summary:**

This paper proposes an approach to synthesizing loop invariants with reinforcement learning. The idea is to leverage abductive reasoning for which search strategies can be treated as non-deterministic programs. An AlphaZero-style RL algorithm is in turn used to refined the strategies. The paper also proposes a method to generate problems of invariant synthesis using RL. To automate some of the processes described in the paper, the authors also implemented the Looprl theorem prover consisting of features that ease the burden of customising the approach. The evaluation clearly shows the benefits of including the teacher process in this approach.

**Questions:**

I would consider raising the scores if the authors weaken the claims, or provide convincing reasons why the approach would work for general automated or interactive theorem proving.

**Limitations:**

Yes, the authors adequately addressed the limitations.

**Strengths And Weaknesses:**

Originality and clarity:

- The idea of using RL for invariant synthesis itself is not new. However, to my knowledge the proposed method to generate new problems (i.e., the teacher process) is indeed novel. Even better, the generation of new problems uses the same methodology used to solve the problems, which leads to a uniform AlphaZero-style solution to the problem of invariant synthesis. The Looprl theorem prover clearly embodies a significant amount of work which makes the proposed approaches practical. The paper is well written and easy to follow. Overall, it is a good attempt to find loop invariants using a combination of abductive reasoning and reinforcement learning.

Quality and significance:

- My main concern is the significance. It is true that abductive reasoning is closer to the way human experts reason about invariants, but the proposed approach is conceivably insufficient for the general task of theorem proving. The crucial part of this approach is the abduct procedure, which requires intensive domain-specific expert knowledge to engineer.

    In the case of loop invariant synthesis studied in this paper, inventing such an abduct procedure is relatively easy because the domain is mostly linear arithmetic and we have the Fourier-Motzkin elimination. However, it is unclear how an abduct procedure can be engineered for more advanced mathematics, say, algebraic geometry. I can imagine that inventing these procedures requires deep analysis of human mathematical activities, which essentially makes the proposed approach relies heavily on hand-engineered procedures.

- For the above reasons, I think that the title and the claims of this paper are perhaps too general. Invariant synthesis is a narrow instance of theorem proving (i.e., generating cut-formulae satisfying certain conditions), and the paper only shows that the approach is promising for invariant synthesis.

- The evaluation confirms the contribution of the teacher process, but is rather unconvincing in terms of absolute performance. The fact that looprl with the vanilla MCTS can solve all the problems of Code2inv benchmarks means that the expert knowledge is perhaps too strong, or that the problems are too easy, possibly both. In fact, a modern state-of-the-art invariant synthesizer such as [Spacer](https://arxiv.org/abs/1405.4028) should also be able to solve all the problems in the Code2inv benchmarks. The reviewer suggests adding more problems to the test set, such as problems from recent [SyGuS competitions](https://sygus.org).

---

> ### Author Response · Authors · 2022-08-02
> **Response to reviewer xxib**
>
> We thank you for an interesting and helpful review. We are glad to read that you appreciated our "uniform AlphaZero-style approach" to generating and solving problems along with our engineering work.
>
> We provide a detailed answer to your question about the relevance of our approach to general theorem proving in our common answer to all reviewers. We address your other questions and technical comments below.
>
> ---
>
> > In the case of loop invariant synthesis studied in this paper, inventing such an abduct procedure is relatively easy because the domain is mostly linear arithmetic and we have the Fourier-Motzkin elimination. However, it is unclear how an abduct procedure can be engineered for more advanced mathematics, say, algebraic geometry.
>
> You are right that designing _deterministic_ abduction procedures is hard and successful attempts at doing so have been confined to specific decidable theories (e.g. theories that admit quantifier elimination, which include real algebraic geometry). However, the focused subproblems on which abduction is typically useful are particularly likely to fall into the realm of such theories. More importantly, our framework allows the abduction procedure itself to be nondeterministic, making it amenable to learning in the exact same way other parts of the main search strategy are.
>
> In our use-case, there is little need for learning an abduction procedure since Fourier-Motzkin elimination already provides a simple and effective answer. This does not have to be the case in general though. Incidentally, we started experimenting with an alternate abduction procedure that relies on a more general rewriting engine and that supports arbitrary axiomatized functions. This procedure leverages nondeterministic choice to annotate its input in such a way to guide the saturation algorithm (e.g. what premises are relevant, what rewriting rules should be used...).
>
> ---
>
> > The fact that looprl with the vanilla MCTS can solve all the problems of Code2inv benchmarks means that the expert knowledge is perhaps too strong, or that the problems are too easy, possibly both.
>
> We address the general question of our choice of the Code2Inv benchmarks in our common answer to all reviewers. To address some of your more specific points:
>
> - You can see how much expert knowledge is given to the solver since a full listing of our solver strategy is available in appendix.
> - Although vanilla MCTS can solve all Code2Inv problems, doing so requires many more simulations than are ever available to AlphaZero during training (>1K vs 32 simulations).
> - With search disabled, the Code2Inv benchmarks are still of suitable difficulty for establishing the impact of learning for both the teacher and solver agents (as you note in your review).

---

### Official Review · Reviewer_Hkk3 · 2022-07-11

**Rating:** 7
**Confidence:** 5
**Soundness:** 3 good
**Presentation:** 3 good
**Contribution:** 3 good

**Summary:**

This paper proposes an AlphaZero-like approach to loop invariant synthesis. Specifically, a teacher agent is first trained to generate programs, which are then used to train a solver agent. To incorporate domain knowledge, both agents are modeled as non-deterministic programs, which can be refined by implementing certain operators. The solver agent is a non-deterministic program with the _choose_ operator to be learned and the _reward_ operator to be designed, while the teacher agent is designed to fill a single-loop program template. In order to generate a diverse set of training programs, a number of teacher constraints are introduced. Different from the standard AlphaZero setting, both agents are trained to not only predict the value of a state but also the associated events. The implementation includes a domain-specific language for writing strategies, a graphical interface for inspecting neural network predictions, and an efficient implementation of AlphaZero algorithm. The evaluation performed on the benchmark from Code2Inv shows that Looprl (this work) significantly outperforms Code2Inv.

**Questions:**


In Figure-2, operation (e.g., abduct) and operator (e.g., choose/reward) are highlighted differently. What are the key differences betweem them?

Can you illustrate the rationality of $r_e min(n_e, m_e)$ in the reward design? What is the difference between _number of occurrences_ ($m_e) and _number of calls_ ($n_e)?

How are the set of teacher constraints designed? Are the tuned for the chosen benchmark? What needs to be done to add a new constraint?

**Limitations:**

The current evaluation is relatively weak because the Code2Inv benchmark is a simple set of small single-loop programs. Evaluations on relatively large programs like the ones used in [SV-COMP](https://sv-comp.sosy-lab.org/2022/index.php) will make this work much stronger.

**Strengths And Weaknesses:**

Strengths:
- although fairly technical, this paper is well-writen -- the motivation is clearly illustrated and the proposed methodology of learning teacher and solver agents looks very promising
- modeling search strategies as non-deterministic programs and then reducing strategy learning into refining programs are very novel
- the implementation (including domain specific language design, graphical ainterface, high-performance MCTS search, etc.) is very solid
- the evaluation shows significant improvement compared with Code2Inv

Weaknesses:
- the title makes a very general claim of finding proofs and theorems, however, only loop invariant synthesis (for single-loop programs) is concerned in the paper. The design of teacher and solver agents are also specialied for the loop invariant synthesis problem. It might be better to adjust the title to reflect what the paper is actually trying to address
- as hinted by CLN2INV [29], the Code2Inv benchmarks are very easy to solve when domain knowledge like templates (either explicit or implict) are used.  Verification tasks from [SV-COMP](https://sv-comp.sosy-lab.org/2022/index.php) would be a much better benchmark to test the full strength of Looprl.

---

> ### Author Response · Authors · 2022-08-02
> **Response to reviewer Hkk3**
>
> We thank you for an interesting and insightful review. Also, we are excited to read that you appreciated the novelty of this work along with its engineering and implementation aspects.
>
> We address your questions about the scoping of our contributions and about our choice of the Code2Inv benchmark in a common answer to all reviewers. We answer some of your more specific questions below.
>
> ---
>
> > In Figure-2, operation (e.g., abduct) and operator (e.g., choose/reward) are highlighted differently. What are the key differences between them?
>
> The "choose" and "reward" operators are highlighted as keywords because they are built-in operators in our strategy DSL. Such operators are not available in most programming languages and can hardly be emulated using standard language constructs. In contrast, "abduct" is a standard function that is definable within our strategy language (with or without using the "choose" and "reward" operators).
>
> ---
>
> > Can you illustrate the rationality of $r_e min(n_e, m_e)$ in the reward design? What is the difference between number of occurrences ($m_e$) and number of calls ($n_e$)?
>
> We write $n_e$ the number of times an event $e$ was triggered. In contrast, $m_e$ is a constant quantity that denotes the _maximal_ number of occurrences of $e$ that can be counted during an episode. Once $n_e \geq m_e$, future occurrences of $e$ are discarded. There are two reasons for incorporating such a bound. First, the number of event occurrences is typically predicted by a neural network and doing so is easier when there is only a finite number of values to consider (in which case we can use a simple softmax head for prediction). Second, the $m_e=1$ case is particularly useful to model events such as constraint violations that can be detected multiple times but should only be penalized once. We added some clarification about this in the paper.
>
> ---
>
> > How are the set of teacher constraints designed? Are they tuned for the chosen benchmark? What needs to be done to add a new constraint?
>
> An important feature of our framework is that it provides an easy and flexible way for experts to provide domain-specific knowledge as nondeterministic strategies. Such strategies can (and should) be evolved and improved incrementally. In our use-case, adding a new constraint only requires adding a few lines of code written in our strategy DSL (after which the teacher can be retrained).
>
> To develop the current teacher strategy, we proceeded as follow. First, we had a quick look at the Code2inv benchmark to see what kinds of program structures and constructs we wanted to support. We came up with an initial strategy based on this. The resulting teacher tended to produce an insufficient variety of samples, thereby causing the solver agent to overfit and perform poorly on Code2inv. This prompted us to add four new constraints (the four last ones in Table 3) along with some stochastic transformations (see Table 5). We also iterated on our teacher strategy a few times to fix a number of undesirable reward hacking behaviors. For example, our initial teacher agent would get away with ignoring the constraint of generating disjunctive invariants by generating simplifiable disjunctions of the form $x \ge c \wedge x \ne c$ (which is equivalent to $x > c$ for integer variables). Crucially, the Looprl framework enabled us to implement and profile these improvements quickly (in a few hours in total if we exclude retraining time).
>
> It would certainly be possible to tune our teacher strategy for Code2Inv specifically. For example, we could add more specialized problem templates in addition to the unique generic template we already use (see Figure 3). However, doing so would be of limited interest since the Code2Inv benchmark can be fully solved through search anyway: our interest is in measuring the contribution of learning rather than scoring as high as possible on Code2inv in absolute terms.

---

### Official Review · Reviewer_grDR · 2022-07-12

**Rating:** 4
**Confidence:** 4
**Soundness:** 2 fair
**Presentation:** 2 fair
**Contribution:** 2 fair

**Summary:**

This work presents a self-supervised method that trains a teacher to generate theorems and a solver to prove them.
The method applies to the special case of single-loop programs and an assertion that requires proof.


**Questions:**

1. Is the model overfitting the generated data?
2. Is there a method that generalizes to different types of problems without intensive augmentation or overfitting?


**Limitations:**

The work does not address the limitation of solving one problem type that is too specific and overfitting.

**Strengths And Weaknesses:**

Strengths:
1. Problem formulation of validation assertion is precise.
2. The method uses vast augmentation to generate data.
3. A high-performance implementation of AlphaZero and search.

Weaknesses:
1. The paper is not clearly written.
2. Limited to a very special case of a single-loop.
3. Doesn't generalize, for example to nested loops and does not scale.

---

> ### Author Response · Authors · 2022-08-02
> **Response to reviewer grDR**
>
> Thank you for your comments and for your time reviewing our submission.
>
> ---
>
> **"The method uses vast augmentation to generate data."**
>
> The term _data augmentation_ is misleading about the role that the teacher agent plays in our framework. In our invariant synthesis use case, there is no starting dataset to be augmented. Rather, a reinforcement learning agent discovers a variety of interesting problems based on a unique template by learning to solve random constraints.
>
> ---
>
> **"The paper is not clearly written."**
>
> Some more detailed feedback on what parts or aspects of the paper you found most problematic would be very useful to us since other reviews did not express readability concerns.
>
> ---
>
> **Our method is "limited to a very special case of a single-loop", "does not generalize to nested loops" and "does not scale".**
>
> See our common response to all reviewers. In particular, we point to [another paper](https://arxiv.org/pdf/1906.11033.pdf) that we cite in our work and which essentially proposes a nondeterministic strategy for verifying imperative programs with nested loops and advanced datatypes such as lists. Adapting such a strategy to work with our framework would not be conceptually difficult, although it certainly raises orthogonal engineering challenges.
>
> ---
>
> **"Is the model overfitting the generated data?"**
>
> Our experiment with the Code2inv benchmarks provides a partial answer here. Indeed, the Code2inv benchmarks are only used for evaluation and never seen during training! The space of single-loop programs with linear arithmetic is pretty large and only 3 of the 126 Code2Inv problems were independently generated by the teacher during training (where problem equivalence is computed modulo variable renaming and modulo the value of numerical constants with absolute value greater than 2). It is therefore clear that meaningful generalization happened.
>
> More generally, the solver's generalization capabilities can only be as good as the teacher's ability to generate relevant and diverse samples. Importantly, our framework enables human experts to improve any given teacher strategy in a flexible and incremental way (see example in our answer to reviewer Hkk3).
>
> ---
>
> **"Is there a method that generalizes to different types of problems without intensive augmentation or overfitting?"**
>
> Training a single large network to serve as an oracle for a variety of teacher and solver strategies targeting different types of problems would likely help with achieving better generalization and sample-efficiency. However, investigating this question is beyond the scope of this paper.

---

### Author Response · Authors · 2022-08-02
**A common response to all reviewers**


---

We thank all reviewers for their helpful comments.

This submission contributes an idea that we believe has generalizable potential: learning theorem proving in a purely self-supervised fashion by co-training a teacher and a solver agent to refine nondeterministic search strategies. In addition, we propose two design principles for writing such strategies: the combination of abduction with nondeterministic choice and the _conditional generative strategy_ pattern. The loop invariant synthesis is merely a demonstrating use-case, which we also use as a running example for pedagogical reasons to make matters more concrete.

---

### Relevance of our contributions beyond invariant synthesis

Although rarely made explicit, nondeterministic programming plays an important role as a structuring principle in general theorem proving. Most interactive theorem provers support specifying custom search tactics in dedicated domain-specific languages with support for backtracking choice [a] and nondeterministic proof search strategies have been proposed in many areas of mathematics and programming [b] [c] [d]. However, pure backtracking search does not scale and so a key open question is how to learn effective search oracles and particularly how to do so in contexts of limited to nonexistent training data.

**Using RL to train search oracles**

We propose reinforcement learning as a general approach to training oracles for nondeterministic proof-search strategies. Our core insight is that nondeterministic programming and RL can be similarly combined to implement _teacher_ agents, thereby allowing truly self-supervised learning.

However, doing so raises an important challenge since training such teacher agents requires resolving a conflict between the two objectives of generating valid yet diverse samples [e]. Our paper proposes a key design principle for implementing teacher strategies that avoids this obstacle using what we call "conditional generative strategies". We again believe this is a general idea with broad applicability. For inequality proving, say, such a strategy may sample a random set of constraints that could be translated into something like "generate a 3-variable inequality whose proof involves a nontrivial combination of the Cauchy-Schwarz and rearrangement theorems".

Finally, we propose a number of engineering contributions for making our framework practical, none of which are specific to invariant synthesis (making Looprl admittedly somewhat ill-named). These include a flexible and fast domain-specific language for writing neural-guided strategies with a Python-facing API and an _event_ abstraction to ease reward engineering.

**Leveraging the power of abductive reasoning**

Our paper suggests another design principle for writing both teacher and solver strategies: combine an abduction procedure with nondeterministic choice. Abductive reasoning intuitively plays a major role in the way humans do mathematics and our paper is not the first to suggest its use in automated theorem proving. However, we argue that abductive reasoning combines particularly well with our proposed framework as neural guidance makes it practical by ranking and filtering abduction candidates. Moreover, the idea of abductive reasoning integrates well with nondeterministic programming as fully deterministic abduction procedures are typically only available for specific decidable theories (see response to Reviewer xxib).

**Relevance of our framework to general interactive theorem proving**

More broadly speaking, we see our paper as a first step towards a more general vision for theorem proving where interactive theorem provers allow users to write teacher and solver strategies for various domains of mathematics and programming in a modular and distributed fashion. A unique neural network is then trained as an oracle for these strategies, allowing better generalization and sample efficiency.

---

> ### Author Response · Authors · 2022-08-02
> **A common response to all reviewers**
>
> ---
>
> ### Choice of the Code2inv benchmarks
>
> The choice of the right benchmarks for demonstrating a novel technique is never as clear as for well-established types of approaches. Code2Inv would be a poor choice if our goal were to demonstrate a new state-of-the-art in invariant synthesis. However, we believe that it is suitable for the role of demonstrating our framework in a _well-contained_ setting while establishing the impact of training on both the solver and the teacher agents.
>
> Targeting more advanced benchmarks such as SV-COMP does not present a major conceptual challenge [b]. However, doing so raises orthogonal engineering challenges that we believe justify leaving such an evaluation to future work. In particular, targeting such benchmarks requires targeting a much wider set of program constructs and reasoning principles while adhering to performance standards that are higher than those of typical theorem provers (running AlphaZero on the kind of hardware that is easily available in academia requires simulating thousands of environment steps per second and per CPU core). Also, the sheer size of many problems in those benchmarks, while inconsequential for traditional solving approaches, presents a challenge for methods based on Transformer-like networks for which inference cost scales as a quadratic function of input size.
>
> ---
>
> ### Suggested paper improvements
>
> In response to the reviewers, we changed some parts of the writing of our paper to better communicate our contributions and their scope. A possible source of misunderstanding lies in our use of an atypical presentation format: instead of introducing our framework in an abstract manner and then illustrate it with an invariant synthesis use-case in a distinct section, we made the pedagogical choice of using the use-case as a running example thread to motivate and introduce our contributions. On the one hand, we feel vindicated for this decision as two reviewers pointed out the clarity and accessibility of our paper. On the other hand, this choice admittedly muddies the water when it comes to understand how broadly applicable each contribution really is. Moreover, our paper was definitely missing a more elaborate conclusion section that would state our contributions explicitly and put them in the context of a more global vision. We added such a section in the updated version of our paper.
>
> Regarding changing the title to make it more specific, we feel conflicted. Indeed, as we argue, this paper is only incidentally about invariant synthesis as an example application (after all invariant synthesis is merely the last sentence of the abstract). For the above reasons the paper targets a more general audience. If, given our additional explanations, the reviewers agree, we would like to keep the more general paper title to more clearly communicate the new paradigm for learning theorem provers that this paper truly is about.
>
> **Changes summary (ranked from most to least important)**:
>
> - Added a more elaborate "Conclusions and Future Work" section.
> - Added details to section 2.4 to better motivate and define the concept of a "conditional generative strategy" in a general setting.
> - Added a more general perspective on how our framework enables scalable abductive reasoning in section 2.2.
> - Added details to better motivate the formula for computing final rewards in section 2.5 (answering Reviewer Hkk3).
> - Added more signposting in the introduction to clarify the role of our invariant synthesis use-case.
> - Added a plain English description of the example solver strategy in Figure 2 for better ease of reading.
>
> ---
>
> ### Footnotes and references
>
> - [a] [Ltac](https://coq.inria.fr/refman/proof-engine/ltac.html) is an example of such a domain-specific language for the Coq theorem prover.
> - [b] The [Ilivna](https://arxiv.org/pdf/1906.11033.pdf) paper proposes a nondeterministic strategy for verifying imperative programs with nested loops and advanced datatypes such as lists.
> - [c] A [workshop poster](https://mathai-iclr.github.io/papers/posters/MATHAI_3_poster.pdf) from Selsam provides rough sketches of nondeterministic strategies for real inequality proving and euclidean geometry.
> - [d] A recent [research program announcement](https://gowers.wordpress.com/2022/04/28/announcing-an-automatic-theorem-proving-project/) from Timothy Gowers is making the bet that most problem-solving skills of working mathematicians can be formalized into non-deterministic programs with limited branching.
> - [e] In prior attempts, we tried to refine nonconditional problem generation strategies using policy-gradient algorithms with entropy regularization to encourage diversity. However, this approach failed as the agent had little incentive to try and generate problems with a high rejection risk.

---

### Meta-Review · Area_Chair_kexP · 2022-08-22

**Recommendation:** Accept
**Confidence:** Less certain

**Metareview:**

This work presents an approach to learning to prove loop-invariant theorems, organized around jointly training teacher and solver models. Reviewers praised its originality and creativity, as well as the quality of the software artifacts produced. Both the ideas and the code could be valuable for the community.

At the same time, there is a consensus (which I agree with) that the work oversells itself by claiming to be a general framework for learning to prove theorems. Might be true that you could in principle apply the framework to other kinds of theorems, but that would have to be shown empirically. Ditto for the claim that this can be applied to program synthesis, which the paper makes in the very first sentence of the abstract.

Given these overreaches, the camera ready version of this paper _needs_ to soften its claims about its broad applicability for theorem proving and program synthesis. The authors also need to change the title so that it has "Loop Invariant" in it or something similar, which they are receptive to in the rebuttal. The paper can talk about these loftier ambitions in the conclusion, but should clearly demarcate the actual extent of the empirical results.

**Award:**

No

---

### Decision · Program_Chairs · 2022-09-14

Accept